# Not All Unlabeled Data are Equal: Learning to Weight Data in Semi-supervised Learning

**Zhongzheng Ren,**\* **Raymond A. Yeh,**\* **Alexander G. Schwing**
University of Illinois at Urbana-Champaign
{zr5, yeh17, aschwing}@illinois.edu

## Abstract

Existing semi-supervised learning (SSL) algorithms use a single weight to balance the loss of labeled and unlabeled examples, *i.e.*, all unlabeled examples are equally weighted. But not all unlabeled data are equal. In this paper we study how to use a different weight for *every* unlabeled example. Manual tuning of all those weights – as done in prior work – is no longer possible. Instead, we adjust those weights via an algorithm based on the influence function, a measure of a model's dependency on one training example. To make the approach efficient, we propose a fast and effective approximation of the influence function. We demonstrate that this technique outperforms state-of-the-art methods on semi-supervised image and language classification tasks.

## 1  Introduction

Unlabeled data helps to reduce the cost of supervised learning, particularly in fields where it is expensive to obtain annotations. For instance, labels for biomedical tasks need to be provided by domain experts, which are expensive to hire. Besides the hiring cost, labeling tasks are often labor intensive, *e.g.*, dense labeling of video data requires to review many frames. Hence, a significant amount of effort has been invested to develop novel semi-supervised learning (SSL) algorithms, *i.e.*, algorithms which utilize both labeled and unlabeled data. See the seminal review (specifically Sec. 1.1.2.) by Chapelle et al. [6] and references therein.

Classical semi-supervised techniques [26, 36, 38, 42] based on expectation-maximization [11, 16] iterate between (a) inferring a label-estimate for the unlabeled portion of the data using the current model and (b) using both labels and label-estimates to update the model. Methods for deep nets have also been explored [20, 29, 32, 33]. More recently, data augmentation techniques are combined with label-estimation for SSL. The key idea is to improve the model via consistency losses which encourage labels to remain identical after augmentation [4, 41].

Formally, the standard SSL setup consists of three datasets: a labeled training set, an unlabeled training set, and a validation set. In practice, SSL algorithms train the model parameters on both the labeled and unlabeled training sets and tune the hyperparameters manually based on the validation set performance. Specifically, a key hyperparameter adjusts the trade-off between labeled and unlabeled data. All aforementioned SSL methods use a *single scalar* for this, *i.e.*, an identical weight is assigned to all unlabeled data points. To obtain good performance, in practice, this weight is carefully tuned using the validation set, and changes over the training iterations [4].

We think not all unlabeled data are equal. For instance, when the label-estimate of an unlabeled example is incorrect, training on that particular label-estimate hurts overall performance. In this case, using a single scalar to weight the labeled and unlabeled data loss term is suboptimal. To address this, we study use of an individual weight *for each* of the unlabeled examples. To facilitate such a

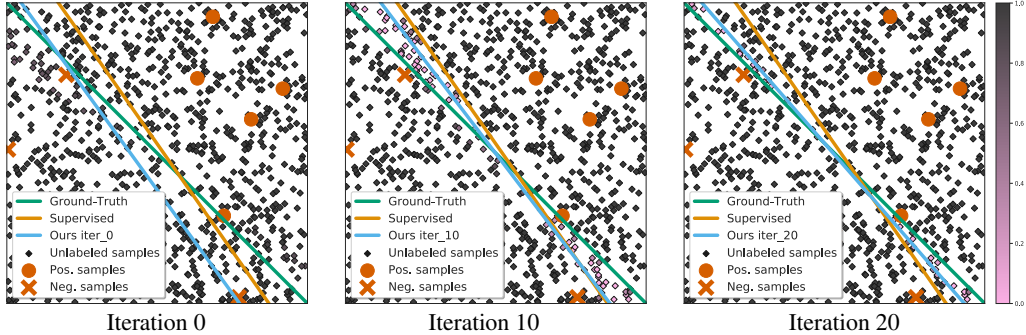

| Iteration 0 | Iteration 10 | Iteration 20 |

Figure 1: Decision boundaries across training iterations on linearly separable data. Labeled samples are shown in **orange** and unlabeled data in **black/pink** (shading depicts weight of each unlabeled point). Our approach **(blue)** with per example weights with Pseudo label SSL algorithm [20].

large number of hyperparameters, we automatically adjust the per-example weights by utilizing the influence function [7]. This influence function estimates the "importance" of each unlabeled example using the validation set performance.

In Fig. 1 we demo this idea on **labeled** and **unlabeled**, linearly separable data. The gray/pink color shade indicates the weight of the unlabeled data. We observe the **proposed method** to more closely mimic **ground-truth** compared to **supervised** training.

The performance gain does not come for free. The method involves adjusting per-example weights for each unlabeled example which is computationally expensive if implemented naively. Specifically, adjusting a per-example weight involves computing (a) a per-example gradient and (b) an inverse Hessian vector product w.r.t. the model parameters. To address both challenges, we design an efficient algorithm for computing per-example gradients, extending backpropagation. Moreover, we propose an effective and efficiently computable approximation specifically for the influence functions of deep nets. These improvements permit to scale the approach to recent SSL tasks and achieve compelling results on CIFAR-10, SVHN, and IMDb.

## 2   Background & Related Work

We first discusss recent advances in semi-supervised learning for image classification, influence functions and gradient based hyperparameter optimization.

**Semi-supervised Learning (SSL).** Given a labeled dataset $\mathcal{D} = \{(x, y)\}$, a set of unlabeled data $\mathcal{U} = \{(u)\}$, and a validation set $\mathcal{V} = \{(x, y)\}$, SSL aims to solve the following program:

$$\min_{\theta} \sum_{(x,y)\in\mathcal{D}} \ell_S(x, y, \theta) + \lambda \sum_{u\in\mathcal{U}} \ell_U(u, \theta), \tag{1}$$

where $\ell_S$ denotes the per-example supervised loss, *e.g.*, cross-entropy for classification, and $\ell_U$ denotes the per-example unsupervised loss, *e.g.*, consistency loss [41] or a regularization term [4, 27]. Lastly, $\theta$ denotes the model parameters and $\lambda \in \mathbb{R}_{\geq 0}$ denotes the scalar weight which balances the supervised and unsupervised loss terms. Note that existing works use a single non-negative real-valued $\lambda$. Tuning of $\lambda$ is performed either manually or via grid-search based on a performance metric assessed on the validation set $\mathcal{V}$.

Different choices of the unsupervised loss $\ell_U$ lead to different SSL algorithms. For example, unsupervised losses $\ell_U(u, \theta)$ resembling a supervised loss $\ell_S$ with the pseudo label $\tilde{y}$, *i.e.*, $\ell_U(u, \theta) \triangleq \ell_S(u, \tilde{y}, \theta)$. In most cases, the pseudo label $\tilde{y}$ is constructed based on the model's predicted probability $p_\theta(k|u)$ for class $k$. The exact construction of the pseudo label $\tilde{y}$ depends on the SSL algorithm.

Specifically, Pseudo-Labeling [20] chooses $\tilde{y}$ to be the label predicted by the current model $p_\theta(k|u)$, *i.e.*, $\tilde{y} = \text{One-Hot}(p_\theta(k|u))$ and uses the cross entropy loss for $\ell_U$. Mean Teacher [40] chooses $\tilde{y}[k] = \sum_i \alpha^i \cdot p_{\theta_i}(k|u)$ to be an exponential moving average of model predictions, where $\alpha$ is a decay factor and $\theta_i$ denotes the model parameters $i$ iterations ago (0 being the most recent). Virtual

Adversarial Training (VAT) [27], MixMatch [4], UDA [41], ReMixMatch [5] and FixMatch [39] all choose the pseudo-labels based on predictions of augmented samples, *i.e.*, $\tilde{y}[k] = p_\theta(k|\text{Augment}(u))$.

For the augmentation Augment($u$), VAT adversely learns an additive transform, MixMatch considers shifts and image flipping, UDA employs cropping and flipping of the unlabeled images, ReMixMatch learns an augmentation policy during training and FixMatch uses a combination of augmentations from ReMixMatch and UDA. In summary, all these methods encourage consistency under different augmentations of the input, which is imposed by learning with the extracted pseudo-label.

Note that all these works use a single scalar weight $\lambda$ to balance the supervised and unsupervised losses. In contrast, we study a per-example weight $\lambda_u$ for each $u \in \mathcal{U}$, as the quality of the pseudo-label varies across unlabeled examples.

**Influence Functions.** Discussed for robust statistics, influence functions measure a model's dependency on a particular training example [7]. More specifically, the influence function computes the change $\frac{\partial \theta^*(\epsilon)}{\partial \epsilon}$ of the optimal model parameters when upweighting the loss of a training example $x$ by a factor $\epsilon > 0$, *i.e.*, $\theta^*(\epsilon) \triangleq \arg\min_\theta \sum_{(x',y') \in \mathcal{D}} \ell_S(x', y') + \epsilon \ell_S(x, y)$. Recently, Koh and Liang [14] utilized influence functions to understand black-box models and to perform dataset poisoning attacks. Moreover, Koh et al. [15] study the accuracy of influence functions when applied on a batch of training examples. Ren et al. [31] use influence functions in the context of robust supervised learning.

Different from these works, we develop an influence function based method for SSL. In the context of hyperparameter optimization, influence functions can be viewed as a special case of a hypergradient, where the hyperparameters are the per-example weights $\lambda_u$. We note that this connection wasn't pointed out by prior works. A review of gradient based hyperparameter optimization is provided next.

**Gradient-based Hyperparameter Optimization.** Gradient based hyperparameter optimization has been explored for decades [3, 18, 21, 22, 24, 37], and is typically formulated as a bi-level optimization problem: the upper-level and lower-level task maximize the performance on the validation and training set respectively. These works differ amongst each other in how the hypergradients are approximated. A summary of these approximations is provided in the Appendix Tab. A1. Theoretical analysis on gradient-based methods for bi-level optimization is also available [8, 13].

In contrast to existing work which tunes *general hyperparameters* such as weight decay, learning rate, *etc.*, we focus on adjusting the per-example weights in the context of SSL. This particular hyperparameter introduces new computational challenges going beyond prior works, *e.g.*, the need for per-example gradients and sparse updates. We address these challenges via an efficient algorithm with a low memory footprint and running time. Thanks to these improvements, we demonstrate compelling results on semi-supervised image and text classification tasks.

## 3 SSL with Per-example Weights

A drawback of the SSL frameworks specified in Eq. (1) is their equal weighting of all unlabeled data via a single hyperparameter $\lambda$: all unlabeled samples are treated equally. Instead, we study use of a different balance term $\lambda_u \in \mathbb{R}_{\geq 0}$ for each unlabeled datapoint $u \in \mathcal{U}$. This permits to adjust individual samples in a more fine-grained manner.

However, these per-example weights introduce a new challenge: manually tuning or grid-search for each $\lambda_u$ is intractable, particularly if the size of the unlabeled dataset is huge. To address this, we develop an algorithm which learns the per-example weights $\lambda_u$ for each unlabeled data point. Formally, we address the following bi-level optimization problem:

$$\min_{\Lambda = \{\lambda_1, ..., \lambda_{|\mathcal{U}|}\}} \mathcal{L}_S(\mathcal{V}, \theta^*(\Lambda)) \text{ s.t. } \theta^*(\Lambda) = \arg\min_\theta \mathcal{L}_S(\mathcal{D}, \theta) + \sum_{u \in \mathcal{U}} \lambda_u \cdot \ell_U(u, \theta), \quad (2)$$

where $\Lambda \in \mathbb{R}_{\geq 0}^{|\mathcal{U}|}$ subsumes $\lambda_u \ \forall u \in \mathcal{U}$ and $\mathcal{L}_S(\cdot, \theta)$ denotes the supervised loss over a labeled dataset, *e.g.*, $\mathcal{L}_S(\mathcal{D}, \theta) \triangleq \sum_{(x,y) \in \mathcal{D}} \ell_S(x, y, \theta)$. Intuitively, the program given in Eq. (2) aims to minimize the supervised loss evaluated on the validation set w.r.t. the weights of unlabeled samples $\Lambda$, while being given model parameters $\theta^*(\Lambda)$ which minimize the overall training loss $\mathcal{L}(\mathcal{D}, \mathcal{U}, \theta, \Lambda) \triangleq \mathcal{L}_S(\mathcal{D}, \theta) + \mathcal{L}_U(\mathcal{U}, \theta, \Lambda)$. Here, $\mathcal{L}_U(\mathcal{U}, \theta, \Lambda)$ denotes the weighted unsupervised loss over the unlabeled dataset, *i.e.*, $\mathcal{L}_U(\mathcal{U}, \theta, \Lambda) \triangleq \sum_{u \in \mathcal{U}} \lambda_u \cdot \ell_U(u, \theta)$.

**Algorithm 1** SSL per-example weight optimization via influence function.

---
1: Initialize model parameters $\theta$, per-example weights $\Lambda$, step size $\eta, \alpha$
2: **while** not converged **do**
3:     **for** $1 \ldots N$ **do**
4:         Sample batches $\mathcal{D}' \subseteq \mathcal{D}, \mathcal{U}' \subseteq \mathcal{U}$
5:         $\theta \leftarrow \theta - \alpha \cdot \nabla_\theta \mathcal{L}(\mathcal{D}', \mathcal{U}', \theta, \Lambda)$
6:     **end for**
7:     Sample batches $\mathcal{D}' \subseteq \mathcal{D}, \mathcal{U}' \subseteq \mathcal{U}, \mathcal{V}' \subseteq \mathcal{V}$
8:     $\theta^* \leftarrow \theta$
9:     Compute gradient $\nabla_\theta \mathcal{L}_U(u, \theta, \lambda_u) \ \forall u \in \mathcal{U}'$
10:     Compute inverse Hessian matrix $H_{\theta^*}^{-1}$
11:     Approximate $\frac{\partial \mathcal{L}_S(\mathcal{V}', \theta^*(\Lambda))}{\partial \lambda_u} \ \forall u \in \mathcal{U}'$ (Eq. 6)
12:     Update per-example weights $\lambda_u \leftarrow \lambda_u - \eta \cdot \frac{\partial \mathcal{L}_S(\mathcal{V}', \theta^*(\Lambda))}{\partial \lambda_u} \ \forall u \in \mathcal{U}'$
13: **end while**

---

When optimization involves deep nets and large datasets, adaptive gradient based methods like Stochastic Gradient Descent (SGD) have shown to be very effective time and again [3, 19]. Here too we use gradient based methods for both the inner and outer optimization. Hence, the algorithm iteratively alternates between updating the model parameters $\theta$ and the per-example weights $\Lambda$, as summarized in Alg. 1. Optimization w.r.t. $\theta$, while holding $\Lambda$ fixed, involves several gradient descent updates on the model parameters $\theta$ to reduce the loss, *i.e.*,

$$\theta \leftarrow \theta - \alpha \cdot \nabla_\theta \mathcal{L}(\mathcal{D}, \mathcal{U}, \theta, \Lambda). \tag{3}$$

Here, $\alpha > 0$ is the step size. After having updated $\theta$, $\Lambda$ is adjusted based on the gradient of the validation loss:

$$\lambda_u \leftarrow \lambda_u - \eta \cdot \frac{\partial \mathcal{L}_S(\mathcal{V}, \theta^*(\lambda))}{\partial \lambda_u} \quad \forall u \in \mathcal{U}, \tag{4}$$

with $\eta > 0$ denoting the step size. These two update steps are performed until the validation loss $\mathcal{L}_S(\mathcal{V}, \theta(\Lambda))$ converges. To compute the updates for $\lambda_u$, we decompose the gradient by applying Danskin's theorem [10]:

$$\frac{\partial \mathcal{L}_S(\mathcal{V}, \theta^*(\Lambda))}{\partial \lambda_u} = \nabla_\theta \mathcal{L}_S(\mathcal{V}, \theta^*(\Lambda))^\top \ \frac{\partial \theta^*(\Lambda)}{\partial \lambda_u} \ \forall u \in \mathcal{U}. \tag{5}$$

Recall that $\theta^*$ is a function resulting from an optimization with dependencies on $\Lambda$. Computing the gradient with respect to $\lambda_u$ hence requires differentiating through the optimization procedure or the program $\arg\min_\theta \mathcal{L}_S(\mathcal{D}, \theta) + \sum_{u \in \mathcal{U}} \lambda_u \cdot \ell_U(u, \theta)$. Several methods have been proposed to approximate $\frac{\partial \theta^*(\Lambda)}{\partial \lambda_u}$ as discussed in Sec. 2.

In practice, we found the approximation from Cook and Weisberg [7] and Koh and Liang [14] to work well. If $\mathcal{L}$ is twice differentiable and has an invertible Hessian, then Eq. (5) can be written as:

$$\frac{\partial \mathcal{L}_S(\mathcal{V}, \theta^*(\Lambda))}{\partial \lambda_u} = -\nabla_\theta \mathcal{L}_S(\mathcal{V}, \theta^*)^\top \ H_{\theta^*}^{-1} \ \nabla_\theta \ell_U(u, \theta^*), \tag{6}$$

with the Hessian $H_{\theta^*} \triangleq \nabla_\theta^2 \mathcal{L}(\mathcal{D}, \mathcal{U}, \theta^*, \Lambda)$. Observe that Eq. (6) measures how up-weighting a training point changes the validation loss, where the derivative $\frac{\partial \theta^*(\Lambda)}{\partial \lambda_u}$ is approximated using influence functions [7].

When using deep nets, computing Eq. (6) for all unlabeled examples is challenging. It requires to evaluate per-example gradients for each unlabeled example ($\nabla_\theta \ell_U(u, \theta^*) \ \forall u \in \mathcal{U}$) and to invert a high dimensional Hessian ($H_{\theta^*}$). Therefore, in the next section, we discuss approximations which we empirically found to be effective when using these techniques for SSL.

### 3.1 Efficient Computation of Influence Approximation

As mentioned before, computing the influence function in Eq. (6) requires addressing two bottlenecks: (a) Computation of per-example gradients (line 9 of Alg. 1); and (b) Computation of the inverse

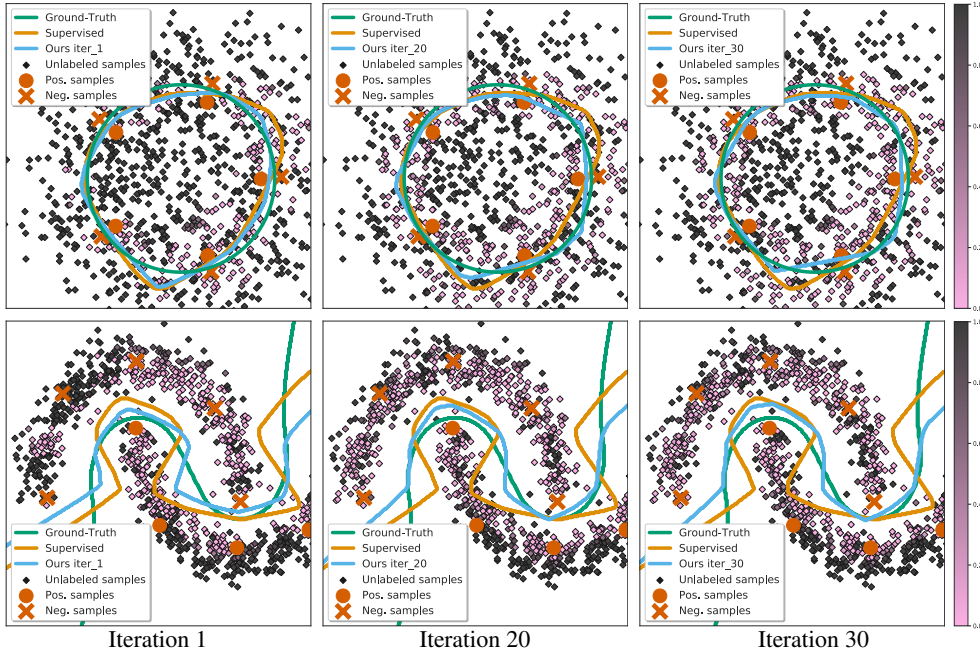

| Iteration 1 | Iteration 20 | Iteration 30 |

Figure 2: The learned decision boundary on the Circles (**Top**) and Moons (**Bottom**) dataset. Visualization scheme follows Fig. 1. Observe the changes in weights and the decision boundary. For example, in the top row, the unlabeled examples near the bottom of the circle are down-weighted at iteration 1, which allows for the decision boundary to shrink towards the ground-truth, at iteration 20.

Hessian (line 10 of Alg. 1). In the remainder of this section, we describe how we tackle both challenges.

**Computation of Per-example Gradient** $\nabla_\theta \mathcal{L}_U(u, \theta)$. Updating $\Lambda$ requires the gradient of the unsupervised training loss $\mathcal{L}_U$ w.r.t. the model parameters $\theta$ individually for each unlabeled point $u \in \mathcal{U}'$. However, backpropagation in deep nets [34] uses mini-batches and stores cumulative statistics rather than an individual example's gradients.

A naive solution applies standard backpropagation to mini-batches containing one example, ideally in parallel. However, this approach remains too slow for our use case. To improve runtime, we leverage the fact that standard auto-differentiation tools for deep nets efficiently compute and store the gradient w.r.t. a layer *activation* $h_u$ for each example $u$. Applying the chain-rule, the per-example gradient w.r.t. the model parameters $\theta$ is then obtained via $\frac{\partial \mathcal{L}_U}{\partial h_u} \cdot \frac{\partial h_u}{\partial \theta}$. Hence, we run standard mini-batch backpropagation to obtain $\frac{\partial \mathcal{L}_U}{\partial h_u}$ for all examples in the mini-batch, followed by parallel computations which multiply with $\frac{\partial h_u}{\partial \theta}$. We describe this approach using a fully connected layer as an example.

Consider a per-example loss $\ell_U(u, \theta) \triangleq \ell(\theta^\top u)$ with a fully connected layer parametrized by $\theta$. Let $h_u \triangleq \theta^\top u$ denote the deep net activation for example $u$. Auto-differentiation tools compute the gradient w.r.t. $h_u$ of the loss $\mathcal{L}_U(\mathcal{U}', \theta) = \sum_{u \in \mathcal{U}'} \ell_U(u, \theta)$ over a mini-batch $\mathcal{U}'$. Due to linearity of gradients, $\frac{\partial \mathcal{L}_U}{\partial h_u} = \frac{\partial \ell_U(u, \theta)}{\partial h_u}$, which is obtained efficiently for all $u \in \mathcal{U}'$ in a single backward pass. Next, observe that the per-example gradients w.r.t. $\theta$ are efficiently computable on a GPU via an element-wise multiplication. Note that standard backpropagation employs an inner product as opposed to an element-wise multiplication. Information about how to compute per-example gradients for other layers is provided in Appendix B.

**Influence Approximation.** A second bottleneck for computing the influence function in Eq. (6) is the inverse Hessian $H_{\theta*}^{-1}$. Directly computing a Hessian for a modern deep net is not practical due to the huge memory footprint. In addition, computing its inverse scales worse than quadratically. While various approximations have been proposed, they are either too slow or not accurate enough for this application as we show in Sec. 4.3.

Most effective in our study was to approximate Eq. (6) by assuming that only the last layer of a deep net is trainable, *i.e.*, we only consider a subset of the parameters $\hat{\theta} \subset \theta$. Exactly computing

| Dataset | CIFAR-10 | | | | | SVHN | | | | |
|---|---|---|---|---|---|---|---|---|---|---|
| # Labeled | 250 | 500 | 1000 | 2000 | 4000 | 250 | 500 | 1000 | 2000 | 4000 |
| Pseudo-Label | 49.98±1.17 | 40.55±1.70 | 30.91±1.73 | 21.96±0.42 | 16.21±0.11 | 21.16±0.88 | 14.35±0.37 | 10.19±0.41 | 7.54±0.27 | 5.71±0.07 |
| VAT | 36.03±2.82 | 26.11±1.52 | 18.64±0.40 | 14.40±0.15 | 11.05±0.31 | 8.41±1.01 | 7.44±0.79 | 5.98±0.21 | 4.85±0.23 | 4.20±0.15 |
| Mean-Teacher | 47.32±4.71 | 42.01±5.86 | 17.32±4.00 | 12.17±0.22 | 10.36±0.25 | 6.45±2.43 | 3.82±0.17 | 3.75±0.10 | 3.51±0.09 | 3.39±0.11 |
| MixMatch | 11.08±0.87 | 9.65±0.94 | 7.75±0.32 | 7.03±0.15 | 6.24±0.06 | 3.78±0.26 | 3.64±0.46 | 3.27±0.31 | 3.04±0.13 | 2.89±0.06 |
| UDA | 8.76±0.90 | 6.68±0.24 | 5.87±0.13 | 5.51±0.21 | 5.29±0.25 | 2.76±0.17 | 2.70±0.09 | 2.55±0.09 | 2.57±0.09 | 2.47±0.15 |
| Re-MixMatch | 6.27±0.34 | - | 5.73±0.16 | - | 5.14±0.04 | 3.10±0.50 | - | 2.83±0.30 | - | 2.42±0.09 |
| FixMatch (CTA) | 5.07±0.33 | - | - | - | 4.31±0.15 | 2.64±0.64 | - | - | - | 2.36±0.19 |
| FixMatch* (CTA) | 5.23±0.28 | - | 4.82±0.09 | - | 4.48±0.15 | 2.77±0.73 | - | 2.41±0.14 | - | 2.17±0.08 |
| Ours (UDA) | 5.53±0.17 | 5.38±0.23 | 5.17±0.16 | 5.14±0.17 | 4.75±0.28 | 2.45±0.08 | 2.39±0.04 | 2.33±0.06 | 2.32±0.06 | 2.35±0.05 |
| Ours (FixMatch, CTA) | 5.05±0.12 | - | 4.68±0.14 | - | 4.35±0.06 | 2.63±0.23 | - | 2.34±0.15 | - | 2.15±0.03 |

Table 1: Test error rate (%) of methods using Wide ResNet-28-2 on CIFAR-10 and SVHN. For our method, we report the mean and standard deviation over 5 runs. (*: reproduced using released code.)

the inverse Hessian w.r.t. $\hat{\theta}$ is reasonably fast as its dimensionality is smaller. Importantly, the per-example gradients discussed in the aforementioned paragraph now only need to be computed for $\hat{\theta}$. Consequently, no backpropagation through the entire deep net is required. In Sec. 4.3 we empirically validate that this method greatly accelerates the training process without a loss in accuracy.

**Efficient Optimizer for $\Lambda$.** In every iteration the discussed approach updates $\lambda_u \ \forall u \in \mathcal{U}' \subseteq \mathcal{U}$, *i.e.*, only a subset of the weights are considered. Intuitively, one might implement this by using a separate optimizer for each $\lambda_u$, *i.e.*, a total of $|\mathcal{U}|$ scalar optimizers. However, this is slow due to the lack of vectorization. To improve, one may consider a single optimizer for $\Lambda$. However, this approach does not perform the correct computation when the optimizer keeps track of statistics from previous iterations, *e.g.*, momentum. Specificallly, the statistics for all dimensions in $\Lambda$ are updated in every step, even if an example *is not* in the sampled subset, which is not desirable.

To get the best of both worlds, we modify the latter approach to only update the subset of $\Lambda$ and their statistics that are selected in the subset $\mathcal{U}'$. We combined this selective update scheme with the Adam optimizer, which we named M(asked)-Adam. For more details see Appendix C.

## 4 Experiments

In this section, we first analyze the effectiveness of our method on low-dimensional datasets before evaluating on standard SSL benchmarks including CIFAR-10 [17], SVHN [28], and IMDb [23]. The method achieves compelling results on all benchmarks. Finally, we ablate different components of the method to illustrate robustness and efficiency. For implementation details, please refer to Appendix D.

### 4.1 Synthetic Experiments

**Datasets and Model.** Beyond the linearly separable data shown in Fig. 1, we consider two additional datasets with non-linear decision boundary, Circles and Moons. The Circle dataset's decision boundary forms a circle, and the Moon dataset's decision boundary has the shape of two half moons, as shown in Fig. 2. Each dataset consists of 10 labeled samples, 30 validation examples[2] and 1000 unlabeled examples. We train a deep net consisting of two fully-connected layers with 100 hidden units followed by a ReLU non-linearity. The models are trained following Alg. 1 using Adam optimizer and using pseudo label [20] as the base SSL algorithm.

**Discussion.** The approach successfully learns models that fit the ground-truth decision boundary on both datasets. As illustrated using colors in Fig. 2, unlabeled examples that are down-weighted the most are near but on the wrong side of the learned decision boundary. This demonstrates that the influence function successfully captures a model's dependency on the training examples. By adjusting the per-example weights on the unlabeled data, the model was able to more closely match the ground-truth.

### 4.2 Semi-supervised Learning Benchmarks

We now evaluate our method using per-sample weights on $\ell_U$ defined by UDA [41] and FixMatch [39].

| Max seq. length | # Labeled | Methods | Error |
|---|---|---|---|
| no truncation | 25,000 | Dai and Le [9] | 7.24 |
| 400 | 25,000 | Miyato et al. [27] | 5.91 |
| 512 | 25,000 | BERT [12] | 4.51 |
| no truncation | 25,000 | Sachan et al. [35] | 4.32 |
| 512 | 20 | UDA [41] | 4.2 |
| 128 | 20 | Supervised | 39.40 |
| 128 | 20 | UDA [41] | 8.98±0.26 |
| 128 | 20 | Ours | **8.51±0.14** |

Table 2: IMDb classification test error rate (%). We report the mean and standard deviation over 3 runs for UDA and our method.

**Image Classification.** Experiments are conducted on CIFAR-10 and SVHN and results are compared to recent methods including Pseudo-Label [20], VAT [27], Mean-Teacher [40], MixMatch [4], UDA [41], ReMixMatch [5], and FixMatch [39]. Following these works, we use Wide-ResNet-28-2 [43] with 1.5M parameters for all experiments for a fair comparison.

We experiment with a varying number of labeled examples from 250 to 4000 and a validation set of size 1024. For completeness we provide in Sec. 4.3 an ablation w.r.t. different validation set sizes, from 64 to 5000. Note that the validation set is *smaller* than that of prior works: MixMatch, Re-MixMatch, and FixMatch use a validation set size of 5000, as specified in their released code. Pseudo-Label, Mean-Teacher, and VAT use a size of 5000 for CIFAR10 and 7000 for SVHN (see Oliver et al. [29]). We use a smaller validation set as we think 5000 validation examples isn't a practical amount: a setting with 250 labeled training samples would result in $20\times$ more validation samples.

SSL benchmark results are provided in Tab. 1. Observe that across different splits the best model outperforms all prior methods achieving improvements over recent baselines like UDA and FixMatch. **For UDA:** the method outperforms the UDA baseline across all splits in both CIFAR-10 and SVHN. **For FixMatch:** we use their best variant of CTAugment and report the numbers from the original paper [5] (See FixMatch (CTA) in Tab. 1). To reproduce the numbers (FixMatch* (CTA) in Tab. 1) we use the released code which seems to result in numbers that differ slightly. Observe that per-example weighting is able to improve upon the original FixMatch baseline results over all splits.

**Text Classification.** We further evaluate the method on language domain data using the IMDb dataset for binary polarity classification. IMDb consist of $25k$ movie reviews for training data and $25k$ for testing. This dataset comes with $50k$ additional unlabeled data and is therefore widely used to evaluate SSL algorithms.

Following the experimental setup of UDA, the model is initialized using parameters from BERT [12] and fine-tuned on IMDb. We use 20 labeled samples for the supervised training set and another 20 for validation. The remaining data is treated as unlabeled.

Note that the maximum sequence length is an important factor in determining the final performance. Normally, the longer the sequence, the better the results. The best result of UDA is achieved using a length of 512 on v3-32 Cloud TPU Pods. However, we mostly have access to 16GB GPUs and very limited access to 32GB GPUs. Due to this hardware constraint, we report results with a maximum sequence length of 128.

The results are shown in Tab. 2, where per-example weights achieve a performance gain over the UDA baseline in the 128 max sequence length setting. For completeness we provide results, with various max sequence lengths from recent SSL approaches in the top half of Tab. 2.

## 4.3 Ablation Studies and Analysis

In this section, we perform numerous ablation studies to confirm the efficacy for each of the components. All the experiments are conducted using CIFAR-10 and the UDA baseline.

**Comparison of Influence Function Approximation.** We compare the method with recent Hessian approximations: Luketina et al. [22] approximate the inverse hessian using an identity matrix, and Lorraine et al. [21] use the Neumann inverse approximation for efficient computation. Note that for Wide-ResNet-28-2 the Neumann approximation requires a large memory footprint as recent SSL algorithms use large batch sizes during training. With a 16GB GPU, we are unable to apply their

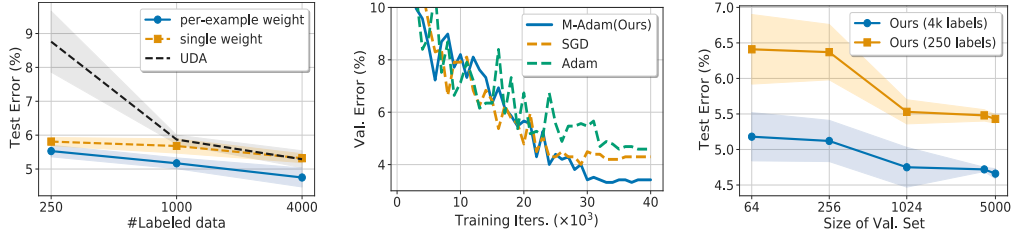

Figure 4: **Left:** Test error comparison between tuning a single weight and per-example weights over different amounts of labeled data. **Center:** Validation error during training of models using different optimizers. **Right:** Test error comparison of models using different validation set sizes. All experiments are conducted on CIFAR-10.

approximation to all the model parameters. To address this, we only apply their approach to the last ResNet block and to the classification layers.

In Fig. 3, we plot the validation error rate over training iterations. In the earlier iterations, the method is on par with Lorraine et al. [21]. In the final iterations, the approach outperforms the baselines. We suspect that the earlier layers in the model have converged, hence, computing the influence based on the exact inverse Hessian of the last layer becomes accurate. In contrast, baselines will continue to compute the influence based on an approximated inverse Hessian. Hence use of the exact inverse leads to better convergence. The improvement on validation performance also transfers to the test set. Ultimately, the method achieves a test error of 4.43%, outperforming 4.51% and 4.85% by Luketina et al. [22] and Lorraine et al. [21], respectively.

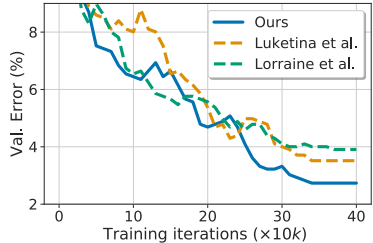

Figure 3: Val. error rate (%) over training iterations for different approximations of the influence function.

**Tuning a Single Weight** $\lambda$**.** To demonstrate the benefits of per-example weights, we perform an ablation study isolating this factor. We apply the method to tuning of a single $\lambda$, shared across all unlabeled examples, following Eq. (1). As shown in Fig. 4 (left), models with per-example weights outperform models with a single $\lambda$ across different data splits. This verifies the hypothesis that not all unlabeled data are equal and that this method can adjust these weights effectively to improve model performance. Average results over three runs are reported.

**Ablation on Adam Implementation.** We demonstrate the effectiveness of the M(asked)-Adam in Fig. 4 (center). We compare with vanilla Adam and SGD. We observe that M-Adam performs the best, followed by SGD, and lastly vanilla Adam. This result highlights the importance of masked updates to correctly compute the running averages of gradient statistics.

**Effect of Validation Size.** As for all SSL algorithms, the validation set size plays an important role for performance. We study the effect of the validation set size on the final performance when using the proposed method. As shown in Fig. 4 (right), results improve consistently from a small validation set (64 samples) to a relatively large one (5000 samples) for both 250 and 4000 labeled data. Average results over three runs are reported.

**Robustness to Hyperparameters** Alg. 1 introduces two hyperparameters: the inner steps $N$ and the step size $\eta$ for tuning $\Lambda$. We study the robustness to these hyperparameters following the UDA setup. Results are shown in Tab. 3. We observe that a large or small $N$ hurts the overall performance. Similarly, the step size $\eta$ for updating $\Lambda$ in the outer loop of Alg. 1 affects the balance between the two updates for $\theta$ and $\lambda$. We found that the sweet spot is reached at ($N = 100, \eta = 0.01$) for CIFAR-10 with 4000 labeled data. We use these hyperparameter values for all splits across the CIFAR-10 and SVHN datasets and found them to work well.

## 4.4 Running Time Comparisons

We provide running time results using Wide-ResNet-28-2 with a batch size of 64, 256, 320 for labeled, unlabeled and validation data respectively. We report the mean running time over 20 iterations.

| $(N, \eta)$ | $(30, 0.01)$ | $(300, 0.01)$ | $(\mathbf{100}, \mathbf{0.01})$ | $(100, 0.1)$ | $(100, 0.001)$ |
|---|---|---|---|---|---|
| Err. | 5.13 | 4.59 | **3.42** | 6.16 | 4.10 |

Table 3: Ablation study on hyperparameters $N, \eta$. We report the val. error rates on CIFAR-10 with 4000 labeled data.

**Per-example Gradient.** We consider two baseline implementations for computing per-example gradients: a *serial* implementation which iterates over each example in the batch, and a *parallel* implementation using `tf.vectorized_map`. The serial implementation requires 18.17s on average for a batch of unlabeled examples to compute the gradients for the entire model. Our method achieves 0.94s, which is 19.3× faster. The *parallel* implementation requires a much larger memory footprint and no longer fits into a 16GB GPU.

**Influence Approximation.** We compare our approximation's running time with Luketina et al. [22] and Lorraine et al. [21]. Our approximation takes 0.455s per batch on average with exact inverse Hessian of the classifier layer, which is comparable to work by Luketina et al. [22] (0.399s) which use an identity matrix as the inverse Hessian. Note that we implemented Luketina et al. [22]'s approximation using our fast per-example gradient implementation, which again verifies its effectiveness and general utility.

When compared to Lorraine et al. [21], the approach is 4.6× faster. Their method iteratively approximates the inverse Hessian vector product. Due to the aforementioned (Sec. 4.3) GPU memory constraint, Lorraine et al. [21]'s approach is implemented only on the last ResNet block and the classification layer, which uses 15.8GB of GPU memory. In contrast, the GPU memory consumption of our approach is only 9GB.

# 5 Conclusion

We demonstrate that use of a per-example weight for each unlabeled example helps to improve existing SSL techniques. In contrast to manual tuning of a single weight for all unlabeled examples, as done in prior work, we study an algorithm which automatically tunes these per-example weights through the use of influence functions. For this, we develop solutions to address the computational bottlenecks when computing the influence functions, *i.e.*, the influence approximation and the per-example gradient computation. These improvements permit to scale to realistic SSL settings and to achieve compelling results on semi-supervised image and text classification benchmarks.

## Broader Impact

We propose a method to improve existing semi-supervised learning (SSL) techniques, *i.e.*, achieving better model performance using a limited amount of labeled data. In general, SSL has a large impact on machine learning applications where labeled data are not widely available, *e.g.*, biomedical data, or applications where labeling is expensive, *e.g.*, dense labeling of videos. While our research focuses on classification benchmarks for SSL, in general, improving SSL techniques will further broaden the scope which machine learning can be applied to.

Due to this we foresee a potential positive social impact from our work. In general, we observe that data are being labeled based on the demand of the users. Consider speech recognition datasets: for common languages large scale corpora exists, *e.g.*, the LibriSpeech ASR corpus [30] contains over 1000 hours of English speech. However, very few datasets exist for rare dialects.

In other words, minority groups may benefit less from progress in machine learning as the datasets are not collected/labeled. We hope that improvements in SSL will make machine learning more accessible and applicable to everyone as it reduces the need for a collection of large scale labeled data.

## Acknowledgments and Disclosure of Funding

This work is supported in part by NSF under Grant No. 1718221, 2008387 and MRI #1725729, NIFA award 2020-67021-32799, UIUC, Samsung, Amazon, 3M, and Cisco Systems Inc. (Gift Award CG 1377144). We thank Cisco for access to the Arcetri cluster. We thank Amazon for EC2 credits. RY is supported by a Google PhD Fellowship. ZR is supported by Yunni & Maxine Pao Memorial Fellowship.

## Footnotes

\*Indicates equal contribution

[2]In SSL literature, the validation set is commonly larger than the training set, *e.g.*, prior works use 5k validation data when there are only 250 labeled samples [29].

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
