[Supplementary Material]

# Appendix

In this appendix we first provide additional background (Sec. A) before detailing more information on per-example gradient computation (Sec. B) and optimizer implementation (Sec. C). We then provide implementation details (Sec. D) and more information about influence functions (Sec. E).

## A    Additional Background

### A.1    Gradient-based Hyperparameter Optimization

| Larsen *et al.* [18] | Conjugate gradients (CG) [25] | Identity [22] |
|---|---|---|
| $\nabla_\theta \mathcal{L}_S(\mathcal{V}) \left[ \frac{\partial \mathcal{L}}{\partial \theta} \frac{\partial \mathcal{L}^\top}{\partial \theta} \right]^{-1}$ | $\arg\min_x \| x H_\theta - \nabla_\theta \mathcal{L}_S(\mathcal{V}) \|$ | $\nabla_\theta \mathcal{L}_S(\mathcal{V}) \left[ I \right]^{-1}$ |
| Stochastic CG [14] | Truncated Unrolled Diff. [37] | Neumann [21] |
| Using [2] | $\nabla_\theta \mathcal{L}_S(\mathcal{V}) \sum_{L<j<i} \left[ \prod_{k<j} I - H_\theta \| w_{i-k} \right]$ | $\nabla_\theta \mathcal{L}_S(\mathcal{V}) \sum_{j<i} \left[ I - \frac{\partial \mathcal{L}_T^2}{\partial\theta\partial\theta^\top} \right]^j$ |

Table A1: A summary of methods to approximate the inverse Hessian vector product $\nabla_\theta \mathcal{L}_S(\mathcal{V}) \, H_\theta^{-1}$ in Eq. (6).

Computing Eq. (6), restated here,

$$\frac{\partial \mathcal{L}_S(\mathcal{V}, \theta^*(\Lambda))}{\partial \lambda_u} = -\nabla_\theta \mathcal{L}_S(\mathcal{V}, \theta^*)^\top \, H_{\theta^*}^{-1} \, \nabla_\theta \ell_U(u, \theta^*),$$

is challenging as it involves an inverse Hessian. When using a deep net, the dimension of the Hessian is potentially in the millions, which demands a lot of memory and computing resources. Prior works, summarized in Tab. A1, have proposed various approximations to mitigate the computational challenges. For example, Luketina et al. [22] propose to use an identity matrix as an approximation of the inverse Hessian, and a recent method by Lorraine et al. [21] uses Neumann series to trade-off computational resources for the quality of the approximation. Different from these approximations, our approach has lower computation time and memory usage for tuning per-example weights. For more details please refer to the ablation studies, specifically Sec. 4.3 in the main paper.

## B    Additional Details for Per-example Gradient Computation

In the main paper, we discussed efficient computation of per-example gradients and presented the details for a fully connected layer. In this section, we will provide the details for two more layers, convolution layers and batch-norm.

**Convolutional Layer.** The convolution layer can be reformulated as a fully-connected layer. Hence, theoretically, we can apply the same implementation. In practice, we found that reshaping to a fully connected layer is slow and memory intensive. Hence, we utilize the auto-vectorizing capability in Tensorflow [1]. More specifically, we slice a convolution layer's activation into mini-batches of size 1 and call the backward function in parallel using `tf.vectorized_map`.

**Batch-norm Layer.** Batch normalization is a special case of a fully-connected layer. The trainable parameters are the scalar weights and bias in the affine transformation. Thus, we can follow the implementation used for a fully connected layer.

## C    Additional Details about Efficient Optimizer for $\Lambda$

We illustrate the efficient implementation for updating $\Lambda$ based on the Adam optimizer in Alg. 2. We named this modified version M(asked)-Adam. Recall, we are updating $\lambda_u \in \Lambda$ only if the loss function $\mathcal{L}$ depends on $u \in \mathcal{U}'$, *i.e.*, when the example is in the sampled mini-batch. Importantly, we do not want to update the running averages of the gradients with 0 for all examples which are *not* in the mini-batch. To do so, we introduce a mask $M \triangleq \mathbf{1}[\nabla_\Lambda \mathcal{L}(\Lambda) \neq 0]$ which indicates whether the gradient w.r.t. a particular $\lambda_u$ is 0. We use $\mathbf{1}[\cdot]$ to denote the indicator function.

---

**Algorithm 2** M-Adam Optimizer. We use $\odot$ to denote element-wise vector multiplication.

---

**Require:** $\alpha \in \mathbb{R}_{>0}$: step size
**Require:** $\beta_1, \beta_2 \in [0,1)$: exponential decay rates for computing running averages of gradient and its square
**Require:** $\epsilon$: a fixed small value
**Require:** $\mathcal{L}(\Lambda)$: A stochastic loss function with parameters $\Lambda$.
 1: Initialize $\Lambda, m, v \in \mathbb{R}^{|\mathcal{U}|}$, $t$ and $\theta_0$
 2: **while** not converged **do**
 3:     $t \leftarrow t + 1$
 4:     $g_t \leftarrow \nabla_\Lambda \mathcal{L}_t(\Lambda_{t-1})$ (Compute gradient w.r.t. to the stochastic loss function)
 5:     $M \leftarrow \mathbf{1}[g_t \neq 0]$ (Obtain mask to block updates, $\mathbf{1}$ denotes the indicator function)
 6:     $m_t \leftarrow m_{t-1} + (\beta_1 - 1) \cdot m_{t-1} \odot M + (1 - \beta_1) \cdot g_t$
 7:     $v_t \leftarrow v_{t-1} + (\beta_2 - 1) \cdot v_{t-1} \odot M + (1 - \beta_2) \cdot g_t \odot g_t$
 8:     $\hat{m}_t \leftarrow m_t / (1 - \beta_1^t)$
 9:     $\hat{v}_t \leftarrow v_t / (1 - \beta_2^t)$
10:     $\Lambda_t \leftarrow \Lambda_{t-1} - \alpha \cdot \hat{m}_t \odot M / (\sqrt{\hat{v}_t} + \epsilon)$
11: **end while**

---

## D   Implementation Details

We follow the setup of UDA [41] and FixMatch [39]. We obtain datasets and model architectures from UDA's and FixMatch's publicly available implementation[3,4].

**Image Classification.** For both UDA and FixMatch, we use the same validation set of size 1024. We use M-Adam with constant step size of $0.01$ as discussed in Sec. C to update $\Lambda$, and SGD with momentum and a step size of $0.03$ is used to optimize $\theta$.

For UDA, we set the training batch sizes for labeled and unlabeled data to 64 and 320. The model is trained for 400k steps. The first 20k iterations are the warm-up stage where only network weights $\theta$ are optimized but not $\Lambda$. We initialize $\lambda_u, \forall u \in \mathcal{U}$, to 5 for training with 250 labeled samples and 1 for the other settings. All experiments are performed on a single NVIDIA V100 16GB GPU. The inner step $N$ is set to 100 and the step size $\eta$ is 0.01.

Following FixMatch, the training batch sizes for labeled and unlabeled data are 64 and $448 = 64 \cdot 7$. The model is trained for 1024 epochs. We initialize $\lambda_u, \forall u \in \mathcal{U}$, to 1 for all experiments. The inner step $N$ is set to 512 and step size $\eta$ is 0.01. Each experiment is performed on two NVIDIA V100 16GB GPUs.

**Text Classification.** Following UDA [41], the same 20 labeled examples are used. We randomly sample another 20 to be part of the validation set as UDA did not provide a validation set. The train and validation set have equal number of examples for each category. We use the same unlabeled data split as UDA, except we exclude the examples used in the validation set. In total, we have 69,972 unlabeled samples. We fine-tune the BERT model for 10k steps with the first 1k iterations being the warm-up phase. The training batch sizes for labeled and unlabeled data are 8 and 32. We use Adam to optimize network weights $\theta$ with learning rate $2 \times 10^{-5}$. M-Adam is used to optimize $\Lambda$ with constant learning rate $0.01$, and we optimize $\Lambda$ once every 5 $\theta$ optimization steps. All experiments for text classification are performed on NVIDIA V100 32GB GPUs. As mentioned in Sec. 4.2, UDA uses v3-32 Cloud TPU Pods which allows to train with larger batch sizes and longer sequence lengths. In our case, the largest memory GPUs which we have access to are the V100 32GB GPUs.

**Reparamterization for Binary Classification.** The text classification task contains two classes and uses cross entropy during training. The provided network architecture of UDA predicts two logits $f_{\theta_1}(x)$ and $f_{\theta_2}(x)$ one for each class given an input $x$. While this over-parametrization doesn't hurt the classification performance, it leads to unstable computation of $H_{\theta^*}^{-1}$, as $\theta_1$ and $\theta_2$ are highly correlated.

To handle this concern, we reparametrize the final classification layer to have parameters $\theta' \triangleq \theta_1 - \theta_2$, and we use the logits $f_{\theta'}(x)$ and $-f_{\theta'}(x)$ in the cross-entropy loss. With this implementation,

we can compute a stable inverse Hessian while obtaining the same training loss of the original parametrization.

# E  Additional Discussion on Influence Functions

Eq. (6) is derived by assuming: (a) the training objective $\mathcal{L}$ is twice-differentiable and strictly convex with respect to $\theta$, and (b) $\theta^*$ has been optimized to global optimality. While these assumptions are violated in context of deep nets, prior works [14, 21] have demonstrate that influence functions remain accurate despite the non-convergence and non-convexity of the model. This finding is also consistent with our experimental results: SSL tasks benefit from tuning the per-example weights via influence functions.

For completeness, we provide a standard derivation of the influence function of $\theta$, $i.e.$, $\frac{\partial \theta^*(\Lambda)}{\partial \lambda_j} = -H_{\theta^*}^{-1} \nabla_\theta \ell_U(j, \theta^*)$ for an unlabeled sample $j$ below.

Recall that $\theta^*$ minimize the loss

$$\mathcal{L}(\mathcal{D}, \mathcal{U}, \theta, \Lambda) = \mathcal{L}_S(\mathcal{D}, \theta) + \sum_{u \in \mathcal{U}} \lambda_u \cdot \ell_U(u, \theta).$$

We assume $\mathcal{L}$ is twice-differentiable and strictly convex w.r.t. $\theta$. Therefore, the Hessian matrix $H_{\theta^*} \triangleq \nabla_\theta^2 \mathcal{L}(\mathcal{D}, \mathcal{U}, \theta^*, \Lambda)$ is positive definite and invertible.

Let's say we increase the weight $\lambda_j$ of unlabeled sample $j$ by a small value $\epsilon$ via $\lambda_j \leftarrow \lambda_j + \epsilon$ and optimize the network using the new weights to optimality. We refer to the new optimal weights as

$$\theta_{\epsilon,j}^* = \arg\min_\theta \mathcal{L}_S(\mathcal{D}, \theta) + \epsilon \ell_U(j, \theta) + \sum_{u \in \mathcal{U}} \lambda_u \cdot \ell_U(u, \theta) = \arg\min_\theta \mathcal{L}(\mathcal{D}, \mathcal{U}, \theta, \Lambda) + \epsilon \ell_U(j, \theta).$$

Since $\theta_{\epsilon,j}^*$ minimizes above equation, we then have the first order optimality conditions:

$$0 = \nabla \mathcal{L}(\mathcal{D}, \mathcal{U}, \theta_{\epsilon,j}^*, \Lambda) + \epsilon \nabla \ell_U(j, \theta_{\epsilon,j}^*).$$

As $\theta_{\epsilon,j}^* \to \theta^*$ when $\epsilon \to 0$, we perform a Taylor expansion of the right-hand side:

$$0 = [\nabla \mathcal{L}(\mathcal{D}, \mathcal{U}, \theta^*, \Lambda) + \epsilon \nabla \ell_U(j, \theta^*)] + [\nabla^2 \mathcal{L}(\mathcal{D}, \mathcal{U}, \theta^*, \Lambda) + \epsilon \nabla^2 \ell_U(j, \theta^*)]\Delta_\epsilon + \mathcal{O}(\|\Delta_\epsilon\|),$$

where the parameter change is denoted by $\Delta_\epsilon \triangleq \theta_{\epsilon,j}^* - \theta^*$, and $\mathcal{O}(\|\Delta_\epsilon\|)$ captures the higher order terms.

Ignoring $\mathcal{O}(\|\Delta_\epsilon\|)$ and solving for $\Delta_\epsilon$, we have:

$$\Delta_\epsilon \approx -[\nabla^2 \mathcal{L}(\mathcal{D}, \mathcal{U}, \theta^*, \Lambda) + \epsilon \nabla^2 \ell_U(j, \theta^*)]^{-1} [\nabla \mathcal{L}(\mathcal{D}, \mathcal{U}, \theta^*, \Lambda) + \epsilon \nabla \ell_U(j, \theta^*)].$$

Recall, $\theta^*$ minimizes $\mathcal{L}$. Consequently, we have $\nabla \mathcal{L}(\mathcal{D}, \mathcal{U}, \theta^*, \Lambda) = 0$. Dropping $\mathcal{O}(\epsilon^2)$ terms, we get

$$\Delta_\epsilon \approx -\nabla^2 \mathcal{L}(\mathcal{D}, \mathcal{U}, \theta^*, \Lambda)^{-1} \nabla \ell_U(j, \theta^*)\epsilon = -H_{\theta^*}^{-1} \nabla \ell_U(j, \theta^*)\epsilon.$$

Finally, following the definition of derivatives,

$$\frac{\partial \theta^*}{\partial \lambda_j} = \frac{\theta_{\epsilon,j}^* - \theta^*}{\lambda_j + \epsilon - \lambda_j}\bigg|_{\epsilon \to 0} = \frac{\partial \Delta_\epsilon}{\partial \epsilon} \approx -H_{\theta^*}^{-1} \nabla \ell_U(j, \theta^*),$$

which concludes derivation of the influence function.

## Footnotes

[3]https://github.com/google-research/uda

[4]https://github.com/google-research/fixmatch