[Reviews · NeurIPS 2020]

Review 1

Summary and Contributions: This paper proposed a SSL method with non-uniform weight for unlabeled samples. To learn the per-sample weight on unlabeled data, a bi-level optimization algorithm is introduced to iteratively update the model parameter and per-sample weight until convergence. Experiments are carried out on synthetic and two real datasets.

Strengths: The motivation for per-sample weighted semi-supervised learning is valid and there is very little prior works on this topic. The per-sample weight is estimated with a bi-level optimization algorithm which is straightforward.

Weaknesses: The model parameter which minimizes the full loss (supervised + unsupervised losses) often does not necessarily minimize the supervised loss part. If this holds, the bi-level optimization second step is not reasonable. The minimizer of \Lambda does not necessarily generalize to either train labeled or test data. The author may want to discuss on this point. How does this algorithm avoid trivial solution? A possible trivial one is setting all \lambda=0 and the second step of bi-level optimization is minimized without any regularization. The author may discuss this point. This may also depends on the initialization. The author should elaborate how \lambda is initialized to avoid trivial solution. Given the marginal improvements, sometimes no improvement at all, over the baseline (CTA), the experiment lacks additional insight. In particular, how the final per-sample weight is distributed? This is further demonstrate the effectiveness of the proposed method.

Correctness: There are some issues with the assumption of minimizing the unsupervised loss corresponding to better generalization. Moreover, there is a trivial solution issue.

Clarity: This paper is well written and easy to follow.

Relation to Prior Work: Related works are properly discussed.

Reproducibility: Yes

Additional Feedback: The author should discuss the assumption and trivial solution issues. The experiment section should be improved as well, in particular, what are the per-sample weights and give insight into how it improves upon uniform weight. After reading the response from authors, I feel the overall weights for unlabeled samples are all close to 1, as shown in the figure in rebuttal. This does not discriminately tell which unlabeled samples are important and which are not because giving 10% higher or lower weight does not change the model too much. But in general, the problem proposed in this paper is valid and it could inspire more following works and I will raise my score to a marginal accept.


Review 2

Summary and Contributions: This paper studies on automatically tuning different weights for different unlabeled samples during training in semi-supervised learning (SSL). The learning of the weights relies on influence function, which can approximate each sample's contribution to the learned model's performance. Because computing influence function is costly on large data, this paper proposed methods of handling its computational bottlenecks. The paper demonstrates the benefits of their SSL per-example weight optimization algorithm via influence function by comparing to different benchmarks, and justifies the components of their algorithm with ablation studies. Two main contributions of this paper are: 1) it proposes a per-example weight optimization algorithm for SSL and empirically demonstrates it can improve SSL performance; 2) it proposes a fast and effective approximation of influence function to make it scalable to realistic SSL settings

Strengths: 1) This paper considers an ignorable case in SSL where not all unlabeled data should be treated equally during training. Though automatic weight tuning for different samples is already studies in supervised learning setting, it is new in SSL context to my best knowledge. Therefore, the motivation is clear and valid. The proposed algorithm is simple and practical and demonstrates benefits compared to different baselines, so the paper worked towards its motivation. 2) To learn the weights for different unlabeled data, this paper connects influence function to the weight tuning in SSL. Influence function is a tool of measuring models' dependency on samples in train set. It is previously mainly studied in the area of model interpretability and robust learning, but not widely applied to tuning sample weights in learning models, probably due to its costly computation. This paper addressed influence function's bottlenecks in their algorithm context by 1) making computing per-sample gradient more efficient via utilizing gradient information from layer activation; 2) approximating the hessian matrix from partial training parameters and samples at each iteration. The experiments of comparing to other approximation strategies show the benefits of their approximation methods, providing new approaches and insights for practical use of influence functions in future applications. 3) This paper demonstrates sound empirical evaluation to justify the proposed algorithm. From synthetic experiments, it presents visually the benefits from weighting differently on unlabeled data; from experiments on real data, it compares their algorithm to competitive benchmarks; from ablation studies, it justifies different components of their methods

Weaknesses: I have mainly two concerns. 1) The paper's per-sample weight tuning method relies on influence function. However, the explanation of the influence function holds only when the upweight (i.e. parameter weights in this paper's context) is very close to 0. Otherwise, the theoretical approximation of each sample's contribution from Taylor's first-order expansion is not valid any more. If that's the case (e.g. weights are far away from 0 during training iteration), the influence function might not reflect each sample's influence correctly and the weights produced might be unreliable. 2) I think in the experiment, some interesting baselines are missing. For example, supervise learning methods with per-sample weight tuning, and SSL with other existing per-sample weight tuning methods from supervised learning studies if possible. Further, in the ablation studies section, as the proposed algorithm can be extended to all training samples rather than limited to only unlabeled sample, it could shed more insights if comparing to applying influence function to all training sample, and comparing to weight tuning only on labeled data while the weights fixed on unlabeled samples. These comparisons can help us understand better the roles played by per-sample weight tuning.

Correctness: The general idea and empirical evaluation is correct and valid. But there are some concerns about some details of the correctness, which are mentioned in the weakness section.

Clarity: This paper is well written. The story is clearly told and developed and the statement is easy to follow. One thing I feel not clear is that in the section of Ablation Studies and Analysis, I am not sure if the experiments are multiple runs or a single run. If it is multiple runes, what's the reported statistics (e.g. average, median, etc). If it is single run, why not make it multiple runs as in the section 4.2?

Relation to Prior Work: To my best knowledge, this paper mentioned important related previous works and discussed how their methods differ and improve from existing ones.

Reproducibility: Yes

Additional Feedback: I hope the authors could defend their ideas or answer my questions about the concerns mentioned in the weakness section. I will consider improve my overall score if satisfied with their response. Some tiny suggestions: In the Figure 1 and Figure 2, the weight color range seems too large so I cannot distinguish well between different samples at least at my machine. Perhaps the authors can consider decrease the color range for the weight resolution so more different colors can be viewed. Also, for all figures and tables, the fonts can be a little bit larger. ------------------------After Authors' Response-------------------------------------------------------------------- The authors' response answers all my main concerns. I am satisfied with all of them except for the proof of equation (6). I thought \lambda worked as \epsilon in the original proof of Koh and Liang's ICML paper, where it must be near 0, but in the appendix F of this paper, the authors use \epsilon as the increment to \lambda. However, in the last line of the authors' proof, I don't think it is correct to say that \frac{\partial \theta}{\partial \lambda} = \frac{\partial \Delta}{\partial \epsilon}. In my opinion, it holds when \epsilon has nothing to do with \lambda, but as \lambda is updated iteratively from the current value, \lambda will be related to \epsilon. Perhaps the effects are tiny and it is still a good approximation, but it is unclear. Besides, I agree with R3 about convergence analysis. Though authors response that similar analysis is done in previous work, I think it is best to include it explicitly for their specific algorithm.


Review 3

Summary and Contributions: This paper studies the problem of semi-supervised learning (SSL) and claims that all unlabeled (U) data should not be treated equally in SSL. Then they propose a learning-to-reweight method to adjust the per-example weights using influence function and provide some fast approximations. Experiments on several benchmarks show their method works comparatively or a bit better than state-of-the-art methods.

Strengths: -The idea of the paper is simple and clear. They argue that not all U data are equal and using the same weight for all of them is suboptimal, so they propose to learn per-example weights instead. -The effort made on fast and empirically effective approximations of calculating influence function is another contribution.

Weaknesses: -My first concern is their motivation. From line 32~34, the authors argue that not all U data are equal, because if the label-estimate of U data is incorrect then training on them will hurt the performance. This is not that convincing. As reviewed in Sec. 2, many other SSL methods do not rely on label-estimate, and I don’t see how this argument hold for those methods, why is this a clear drawback? -Noticing that in Tabel 1, the proposed per-example weight method performs much better than label-estimate based SSL method (Pseudo-Label), but has less or no improvement on consistency based SSL method (UDA, FixMatch), which somehow justifies my concern. Could the authors explain more on this?

Correctness: In Figure 4, the per-example weight method has only limited improvement on single weight method. But in Table 1, the best single weight method can achieve almost the same performance as the per-example weight method. What was the single weight method here?

Clarity: The problem of SSL widely exists in the real world and has been a hot research area in recent years. This paper studies it in the different point of view, not all U data are equal, the idea is interesting but there are some unclear parts in the paper that can be further clarified and improved.

Relation to Prior Work: The bi-level optimization problem in Eq 2 seems a bit similar to Eq 1~2 in Reweight [Ren et al., ICML’18], which learns weights of noisy training examples (U data here) by minimize a validation risk. This paper proposes a different approximation method and is shown to work well empirically. But is this algorithm guaranteed to converge under some assumptions? A convergence analysis would be very helpful.

Reproducibility: Yes

Additional Feedback: After rebuttal: I have read the rebuttal and all the reviews. The authors have addressed most of my questions in the rebuttal, so I choose to raise my score.

[Author Response · NeurIPS 2020]

**General response:** We thank all reviewers for their constructive comments. We think there are some **misunderstandings** and we encourage the reviewers to kindly consider our response.

──────────────────── **Reviewer # 1** ────────────────────

**Q1: Parameter minimizing the full loss may not minimize the supervised loss.** Note, "supervised loss" can be computed on the "training" or "validation" set, *i.e.*, $\mathcal{L}_S(\mathcal{D}, \theta)$ or $\mathcal{L}_S(\mathcal{V}, \theta)$. In the constraint, the program optimizes for the optimal model parameters $\theta^*(\Lambda)$ given a particular $\Lambda$, while using the "training" set's supervised loss. In the main objective, the program optimizes for $\Lambda$ based on the supervised loss of the "validation" set. Note the difference in datasets. Intuitively, we look for the optimal parameters on the training set while considering the validation set result. This formalizes manual tuning of hyperparameters in prior work. Hence the bi-level optimization is reasonable.

**Q2: Trivial solution $\Lambda = 0$?** This is a **misunderstanding**, $\Lambda = 0$ is not a trivial solution. $\Lambda = 0$ corresponds to not using any unlabeled data. Due to few labeled data the model achieves error rate ($\downarrow$ is better) of 20.26% on CIFAR10 (4k) and 12.83% on SVHN (1k), much worse than SSL methods [29]. For more empirical evidence we initialize $\Lambda = 0$ and observe our approach to quickly deviate to a non-zero $\Lambda$ (due to high $\mathcal{L}_S(\mathcal{V}, \theta)$). Theoretically, $\Lambda = 0$ is also not a trivial solution: Keep in mind that the upper-level optimizes $\mathcal{L}_S(\mathcal{V}, \theta)$, and the lower-level optimizes $\mathcal{L}_S(\mathcal{D}, \theta)$.

**Q3: Minimize the unsupervised loss ... better generalization?** SSL typically uses an 'unsupervised loss' to leverage unlabeled data. While the model may not generalize if the unsupervised loss is poorly designed, recent works [38, 36] empirically validate their proposed loss. Theoretical analysis of SSL has also been provided under various assumptions, *e.g.*, [6, A]. We encourage R1 to study these works which show how unsupervised losses aid generalization. Such a discussion is beyond the scope of this work. Reference: [A] P. Rigollet, "Generalization Error Bounds in Semi-supervised Classification Under the Cluster Assumption," JMLR 2007

**Q4: How are the final per-sample weights distributed?** Visualizations of per-example weights are color-coded in Fig. 1-2 (paper). Their evolution is shown in the appendix. Notably, unlabeled data with incorrect pseudo-labels are down-weighted near the decision boundary. We also visualize a histogram of per-example weights on CIFAR-10 in the figure to the right.

**Q5: Comparison to uniform weights.** We compare with baselines using uniform weights: (a) In Tab. 1, UDA and FixMatch baselines use uniform weights for all unlabeled data. (b) In Fig. 4 left (orange) we report an ablation study on UDA using a uniform weight learned by our algorithm.

──────────────────── **Reviewer # 2** ────────────────────

**Q6: Influence function holds only when parameter weights are close to 0.** This seems to be a **misunderstanding**. Please see Appendix F. We consider $\lambda_j + \epsilon$ where $\lambda_j$ is the current weight for unlabeled data $j$ and $\epsilon$ is its change. The Taylor expansion holds for any $\lambda_j > 0$ as long as $\epsilon \to 0$.

**Q7: Supervised learning with $\Lambda$ tuning baseline.** We run FixMatch on *labeled* CIFAR-10 (4k) and SVHN (1k) data with per-example weights tuned using our method. We achieve error rates ($\downarrow$ is better) of 13.12% and 8.04% respectively, which are much worse than the results in Tab. 1. This happens as all the unlabeled data is ignored.

**Q8: Per-example weights for both labeled and unlabeled data.** We run FixMatch on CIFAR-10 (4k) and SVHN (1k) with per-example weights for *both labeled and unlabeled data* tuned using our method. We achieve 4.39%, 2.13% error rate respectively, which is on par with Tab. 1 results but not significantly better. This happens as labeled data only occupy a small portion and their labels are clean. Hence, tuning weights for labeled data is not always necessary.

**Q9: Multiple runs for ablation study?** The ablations in Fig. 4 are single run. We run experiments for Fig. 4 left (orange line) 3 times. The mean/std for 250/1k/4k split are 5.81±0.12, 5.68±0.21, and 5.32±0.13, which does not change the conclusions from the ablation study. We'll add multiple runs for all ablations.

──────────────────── **Reviewer # 3** ────────────────────

**Q10: Motivation of per-example weights.** Recent SSL works use pseudo-labels, either hard labels or soft target distributions. This is also true for "consistency based methods" (UDA, FixMatch) where each prediction is a pseudo-label for another example, as explained in L68. Label-estimates are inevitably wrong during training. We introduce per-example weights to handle this. Please see Fig. 1-2 and appendix's animations for a qualitative motivation.

**Q11: Consistency-based SSL methods, or methods without label-estimate.** It's a **misunderstanding** that our method is limited to pseudo-labels. Our approach can work with any unsupervised loss. For some unlabeled data, the unsupervised loss may hurt the validation performance, hence it is beneficial to introduce per-example weights.

**Q12: Improvements on consistency based SSL method (UDA, FixMatch).** "Consistency based methods" also involve label-estimates (soft target). Tab. 1 shows that our method improves the results in 15 out of 16 settings (except for CIFAR-10 with 4k labeled data). This is reasonable as our algorithm will become less beneficial when abundant labeled data is available. While we did not improve upon the published number of FixMatch, our number is still better than the FixMatch results reproduced using *the publicly available code*.

**Q13: Improvements over single-weight baseline (Fig. 4 & Tab. 1).** The black line in Fig. 4 left is the vanilla UDA baseline as in Tab. 1. The orange line in Fig. 4 left is our ablation: a single shared weight $\lambda$ for all examples is learned using the program in Eq. (2). As can be seen, using per-example weights improves over a single weight.

**Q14: Convergence guarantee of bi-level optimization.** As discussed in L95-96, prior works have studied convergence of bi-level optimization. Details are beyond the scope.

[Meta-Review · NeurIPS 2020]

After rebuttal, 2 reviewers initially inclined to reject the paper raise their grades, leading to a consensus among reviewers for weak acceptance. Although their remains room for improvements, the AC agrees that the contributions are solid and interesting for the community, and therefore recommends acceptance. The authors are highly encouraged to update the final version of the paper based on reviewers' comments.